# Decreasing prevalence of contamination with extended-spectrum beta-lactamase-producing *Enterobacteriaceae* (ESBL-E) in retail chicken meat in the Netherlands

**Pepijn Huizinga**[1,2]*, **Marjolein Kluytmans-van den Bergh**[1,3,4], **John W. Rossen**[5], **Ina Willemsen**[1], **Carlo Verhulst**[1], **Paul H. M. Savelkoul**[6,7], **Alexander W. Friedrich**[5], **Silvia García-Cobos**[5¤], **Jan Kluytmans**[1,4]

1 Department of Infection Control, Amphia Hospital, Breda, the Netherlands, 2 Laboratory for Medical Microbiology and Immunology, Elisabeth-TweeSteden Hospital, Tilburg, the Netherlands, 3 Amphia Academy Infectious Disease Foundation, Amphia Hospital, Breda, the Netherlands, 4 Julius Center for Health Sciences and Primary Care, University Medical Center Utrecht, Utrecht University, Utrecht, the Netherlands, 5 University of Groningen, University Medical Center Groningen, Department of Medical Microbiology and Infection Prevention, Groningen, the Netherlands, 6 Maastricht University Medical Center, Caphri School for Public Health and Primary Care, Department of Medical Microbiology, Maastricht, the Netherlands, 7 Amsterdam University Medical Center, Vrije Universiteit Amsterdam, Department of Medical Microbiology & Infection Control, Amsterdam, the Netherlands

¤ Current address: Reference and Research Laboratory on Antimicrobial Resistance and Healthcare-associated infections, National Microbiology Center, Institute of Health Carlos III, Majadahonda, Madrid, Spain

* pepijnhuizinga@gmail.com

**Data Availability Statement:** All raw sequencing reads are available from the ENA database

## Abstract

Retail chicken meat is a potential source of extended-spectrum beta-lactamase-producing *Enterobacteriaceae* (ESBL-E). In the past decade, vast national efforts were undertaken to decrease the antibiotic use in the veterinary sector, resulting in a 58% decrease in antibiotic sales in the sector between 2009 and 2014. This decrease in antibiotic use was followed by a decrease in ESBL-E prevalence in broilers. The current study investigates the prevalence of contamination with ESBL-E in retail chicken meat purchased in the Netherlands between December 2013 and August 2015. It looks at associations between the prevalence of contamination with ESBL-E and sample characteristics such as method of farming (free-range or conventional), supermarket chain of purchase and year of purchase. In the current study, 352 chicken meat samples were investigated for the presence of ESBL-E using selective culture methods. Six samples were excluded due to missing isolates or problems obtaining a good quality sequence leaving 346 samples for further analyses. Of these 346 samples, 188 (54.3%) were positive for ESBL-E, yielding 216 ESBL-E isolates (*Escherichia coli* (n = 204), *Klebsiella pneumoniae* (n = 11) and *Escherichia fergusonii* (n = 1)). All ESBL-E isolates were analysed using whole-genome sequencing. The prevalence of contamination with ESBL-E in retail chicken meat decreased from 68.3% in 2014 to 44.6% in 2015, absolute risk difference 23.7% (95% confidence interval (CI): 12.6% - 34.1%). The ESBL-E prevalence was lower in free-range chicken meat (36.4%) compared with conventional chicken meat (61.5%), absolute risk difference 25.2% (95% CI: 12.9% - 36.5%). The prevalence of

(accession number PRJEB33495). Supermarket chains of purchase have been anonymized.

**Funding:** Part of the sequence data were generated and reused from the Dutch Food & Nutrition Delta Programme 2013 project PathoDetect. Received by: PS. This study was also partially funded by the Amphia Hospital and the medical specialist company Amphia (MSB-A), Breda, the Netherlands. Received by: PH. This study was also partly supported by the Interreg IVa-funded projects "EurSafety Health-Net - EURegional network for patient safety and protection against infections" and "SafeGuard", as part of a Dutch–German cross-border network supported by the European Commission, German Ministry of Economic Affairs, Energy and Industry of the State of North Rhine- Westphalia, and German Ministry for Economic Affairs, Labour, Transport and Digitalisation of the State of Lower Saxony, and the Dutch Provinces of Overijssel, Gelderland, and Limburg. Received by: JR, AWF. The funders had no role in study design, data collection and analysis, decision to publish, or preparation of the manuscript.

**Competing interests:** The authors have declared that no competing interests exist.

contamination with ESBL-E varied between supermarket chains, the highest prevalence of contamination was found in supermarket chain 4 (76.5%) and the lowest in supermarket chain 1 (37.8%). Pairwise isolate comparisons using whole-genome multilocus sequence typing (wgMLST) showed that clustering of isolates occurs more frequently within supermarket chains than between supermarket chains. In conclusion, the prevalence of contamination with ESBL-E in retail chicken in the Netherlands decreased over time; nevertheless, it remains substantial and as such a potential source for ESBL-E in humans.

## Introduction

Infections with extended-spectrum beta-lactamase-producing *Enterobacteriaceae* (ESBL-E) are associated with substantial morbidity, mortality and increased costs compared to infections with their susceptible counterparts [1–4]. Carriage with ESBL-E is often a prerequisite and a predictor for infections with ESBL-E [5–7]. In 2015, about 5% of invasive *E. coli* isolates (blood- and cerebrospinal-fluid cultures) were resistant to third generation cephalosporins in the Netherlands [8]. This is lower than the European population-weighted mean of 13%. Nevertheless, it is more than a five-fold increase since the turn of the century [9].

Originally ESBL-E infections were mainly a hospital-related problem with acquisition in hospitals or related to healthcare contact. This has changed in the past two decades with people that have had no healthcare contact also being rectal carriers of ESBL-E [10,11]. Research efforts have focussed on uncovering routes of transmission and reservoirs of antimicrobial resistant microorganisms and resistance genes by using a one-health approach that includes humans, animals and the environment as an interconnected entity.

Contaminated food has been suggested as a potential source for ESBL-E. ESBL-E contamination rates up to 80% were reported for retail chicken meat in the Netherlands between 2008 and 2010 [12–14]. Exchange of bacteria or genetic material between animals and humans has been suggested, for instance between farmers and their animals, where the epidemiological link is relatively concrete [15]. Transfer of ESBL-E isolates or plasmids carrying resistance genes from bacteria on retail chicken meat to humans in the general community is more difficult to prove due to larger spatial-temporal differences. However, it was recently described that poultry meat can act as a vehicle for exposure and infection with a specific ST131 sublineage [16]. Overlap in genetic content between animal and human domains has clearly been shown, albeit without directionality of possible transmission [13,14,17–19].

Based on the hypothesis of spread of antimicrobial resistant bacteria from chicken meat to humans, the Dutch government set goals to decrease the antibiotic use in Dutch livestock. This initiative resulted in a decrease in antibiotic sales in veterinary medicine of 63.4% between 2009 and 2017 with little to no impact on the production or economic results in the sector [20,21]. Although a causal relation is difficult to prove, the decrease in antibiotic use was followed by a subsequent decrease in isolation of ESBL-E from livestock in the Netherlands. The level of cefotaxime resistance in randomly picked *E. coli* isolates from broiler faeces decreased from 15–20% in 2007 to 1.7% in 2017 [20]. The current study focusses on fresh retail chicken meat from common supermarket chains in the Netherlands. Poultry meat makes up 29% (22kg) of the total meat consumption of the average citizen of the Netherlands and more than half of this consists of chicken breast fillet [22].

The aim of this study is to describe the ESBL-E prevalence in Dutch retail chicken meat over time and in relation to the method of farming (free-range or conventional), and the

supermarket chain where the meat was purchased. In addition, the genetic constitution of the isolated ESBL-E is described.

## Methods

### Sample collection

A convenience sample of chicken meat samples were collected from December 2013 until October 2014 and will be referred to as "period 2014", and from June 2015 until August 2015 this will be referred to as "period 2015". Only unprocessed, raw, conventional or free-range farming chicken-breast fillet was used for this study. Organic chicken meat was not sampled for this study. Only one sample per supermarket chain per day with the same method of farming and/or batch number was included.

The following information was noted for each sample: date of purchase, best before date, supermarket chain and method of farming. Two supermarket chains, which were already part of the same group of supermarkets, merged during the study period and were analysed together as one supermarket chain as we assumed overlapping suppliers already before the official merger. The combined market share of the sampled supermarket chains in the Netherlands is around 70% [23].

The sample size of the second period was calculated after the first collection period. To detect a decline of 15% in ESBL-E prevalence with a power of 80% and an alpha of 0.05 with 142 samples in the first period and an ESBL-E prevalence of around 68% in that first period, 199 samples had to be collected in the second sampling period.

### Microbiological methods

Twelve grams of chicken meat per sample was pre-enriched in 15mL tryptic soy broth (TSB). After overnight incubation, 100 μL of the TSB was transferred to 5mL of selective TSB, containing vancomycin (8 mg/L) and cefotaxime (0.25 mg/L) (TSB-VC). After a second overnight incubation, 10 μL of the TSB-VC was subcultured on an ESBL screening agar, EbSA (AlphaOmega, 's-Gravenhage, the Netherlands), consisting of a split McConkey agar plate containing cloxacillin (400 mg/L), vancomycin (64 mg/L) and on one half cefotaxime (1 mg/L) and the other half ceftazidime (1 mg/L). Species identification (VITEK-MS, bioMérieux, Marcy l'Etoile, France) and antibiotic susceptibility testing (VITEK2, bioMérieux, Marcy l'Etoile, France) were performed for all oxidase-negative Gram-negative isolates that grew on the EbSA agar plate with different morphology. Minimal inhibitory concentrations (MIC) are given in mg/L. The production of ESBL was phenotypically confirmed with the combination disk diffusion method using cefotaxime (30 μg), ceftazidime (30 μg) and cefepime (30 μg) disks, with and without clavulanic acid (10 μg) (Rosco, Taastrup, Denmark). Test results were considered positive if the diameter of the inhibition zone was ≥5 mm larger for the disk with clavulanic acid as compared with the disk without clavulanic acid [24,25]. Antimicrobial susceptibility testing results were interpreted using EUCAST clinical breakpoints (v 7.1) [26].

### Whole-genome sequencing and quality control

All isolates for which ESBL production was phenotypically confirmed were sequenced. Genomic DNA was prepared using the Nextera XT library preparation kit (Illumina, San Diego, United States). The libraries were sequenced on a MiSeq sequencer (Illumina, San Diego, United States) generating 250- to 300-bp paired-end reads using the MiSeq reagent kit v2 or v3 respectively. Quality trimming and *de novo* assembly was performed using CLC Genomics Workbench version 11.0 (Qiagen, Hilden, Germany) as previously described [27]. The

following quality control parameters were considered to assess assembly quality: coverage $\geq 20$; number of scaffolds $\leq 1000$; N50 $\geq 15,000$ bases and maximum scaffold length $\geq 50,000$ bases. If one or more of the criteria was not met, the assembly was excluded from the analyses. In addition, isolates for which the genotypic genus identification did not match the phenotypic (MALDI-TOF) identification were excluded from the analysis.

### Definition of ESBL-E positive samples and isolate selection

Samples were classified as ESBL-E positive when one or more isolates from a sample had a sequence satisfying the quality control criteria and an ESBL gene was located in the sequence data. Samples containing only isolates phenotypically suspected for ESBL production with a good quality sequence where no ESBL gene was identified were reported as ESBL-E negative. Samples were excluded when the only isolate from that sample was phenotypically suspected for ESBL production but sequence data did not satisfy the quality control criteria and hence, no conclusion could be drawn on the on the presence or absence of the ESBL gene.

If samples contained multiple isolates and these clustered according to whole genome multilocus sequence typing and the ESBL gene(s) were identical, only one of the isolates was kept for further analyses.

### Bioinformatics analyses of whole genome sequence data: Species identification, resistance gene detection, plasmid replicon detection, multilocus sequence typing (MLST) and whole-genome MLST (wgMLST)

Assembled genomes were analysed using the bacterial analysis pipeline-batch upload mode from Center for Genomic Epidemiology (accessed week 52 of 2017) (https://cge.cbs.dtu.dk/services/cge/, DTU, Copenhagen) with KmerFinder-2.1 for species identification, ResFinder-2.1 for detection of acquired resistance genes and PlasmidFinder-1.2 for detection of plasmid replicons [28–31]. If multiple plasmid replicons from the same family were detected in one isolate, the plasmid replicon family was counted once for that isolate.

MLST sequence type (ST)(Achtman) was determined using the bioinformatics tool "mlst" by T. Seemann v2.16.1 (https://github.com/tseemann/mlst) [32,33]. For *E. coli* isolates with unknown STs or problems in determining the ST, the raw FASTQ files were submitted to the EnteroBase website to assign new STs (https://enterobase.warwick.ac.uk/species/ecoli v1.1.2) [34].

The phylogroups as described by Clermont *et al* were determined using the ClermonTyping tool v1.0.0 (https://github.com/A-BN/ClermonTyping) which uses a method with different *in-silico* PCR assays and a method using the Mash genome clustering tool [35,36]. When discrepancies between the *in-silico* PCR assay method and the Mash genome-clustering tool method were observed, the phylogroup was reported as "undetermined". The ClermonTyping tool also discriminates between *E. coli* and *E. fergusonii* on the basis of the *citP* gene. If the ClermonTyper identified an isolate as *E. fergusonii* that was previously identified as an *E. coli* (MALDI-TOF and Kmer-Finder 1.2), the species was changed to *E. fergusonii*.

wgMLST analysis was performed for all *E. coli* isolates using Ridom SeqSphere + v5.1.0 (Ridom, Münster, Germany) applying the *E. coli* scheme and clonality threshold according to Kluytmans-van den Bergh *et al*. [27]. Pairwise genetic distances were determined by calculating the proportion of allele differences between isolates. Only good targets present in both sequences were used, ignoring missing values. The threshold used for clonality for *E. coli* was 0.0095 [27]. As a sensitivity analysis the threshold for clonality was doubled to 0.019. Another option to make the criteria for clonality less stringent was taking the core-genome MLST scheme. It was chosen to maximize discriminatory power and work with the originally

proposed cut-off for the wgMLST scheme. A neighbour-joining tree was constructed in Ridom SeqSphere + v5.1.0 using the pairwise genetic distances and metadata were added in the webtool "Interactive Tree of Life" v4.4.2 (https://itol.embl.de)[37–39].

## Statistical analyses

Confidence Intervals (CIs) of proportions were calculated using the adjusted Wald method [40]. All analyses on the ESBL-E prevalence data were performed using Statistical Package for Social Science software (IBM SPSS Statistics 25.0, Armonk, NY). Relative risks for ESBL-E contamination of meat samples were estimated using univariable and multivariable generalized linear models (GLM) with a Poisson distribution, log link and robust error estimation, with year of purchase, supermarket chain and method of farming as independent variables. Associations were measured using relative risks (RR) for a more appropriate interpretation, the high ESBL-E prevalence would lead to high odds ratios overestimating the actual RR [41–43].

Relative risks for clonality were estimated using univariable and multivariable GLM with a binomial distribution, a log link and robust error estimation, with time interval between dates of purchase, supermarket chain (within or between) and farming method (within or between) as independent variables. Due to the non-linear effect of time between isolates related to the frequency of clonality of the pairwise isolate comparisons, it was not suitable as a continuous variable in the logistic regression analyses. As such, time between isolates was taken as a categorical variable with three groups: 0–6 months, 6–12 months and >12 months. The categories were chosen to coincide with changing frequency of clonality, and as such were based on the observed results. As these choices were made with prior knowledge of the data, two alternative models were made excluding the time variable and using shorter time intervals in the first year.

## Accession number

Raw sequencing reads were submitted to the European Nucleotide Archive of the European Bioinformatics Institute and are available under the study accession number PRJEB33495.

## Results

### ESBL-E prevalence survey of retail chicken meat

Of 352 cultures of retail chicken meat six were excluded from further analyses, leaving 346 samples for further analyses, Fig 1. The number of samples taken per month, per supermarket chain and per method of farming is shown in S1 Table. Of the 346 samples, 188 (54.3%) were positive for ESBL-E. Year of purchase, supermarket chain and method of farming were independently associated with the prevalence of contamination with ESBL-E, Table 1. The prevalence of contamination with ESBL-E decreased from 68.3% in the period 2014 to 44.6% in the period 2015, absolute risk difference 23.7% (95% CI: 12.6% - 34.1%) or adjusted relative risk of 0.69 (95% CI: 0.58–0.83) and is shown in more detail in S1 Fig. The prevalence of contamination with ESBL-E was lower in free-range chicken meat (36.4%) compared with conventional chicken meat (61.5%), absolute risk difference 25.2% (95% CI: 12.9% - 36.5%) or adjusted relative risk of 0.60 (95% CI 0.46–0.78), Table 1 and Fig 2. The prevalence of contamination with ESBL-E varied between supermarket chains; the highest ESBL-E prevalence was found in supermarket chain 4 (76.5%) and the lowest in supermarket chain 1 (37.8%).

**Phenotypic and genetic characterization of ESBL-E isolates from retail chicken meat.** A total of 240 isolates were selected for sequencing, 24 were excluded from further analyses for

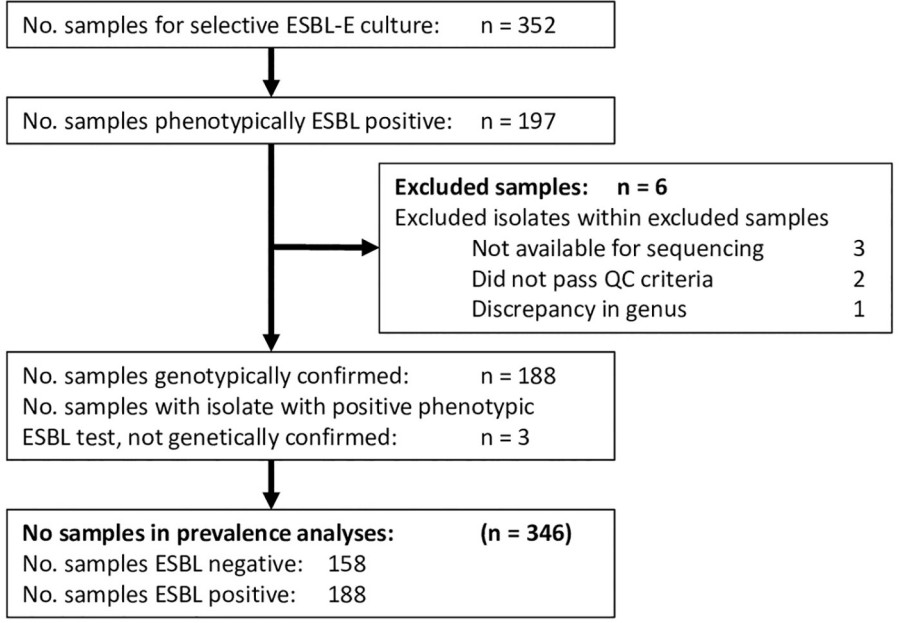

**Fig 1. Flowchart showing the number of chicken meat samples in the study.**

the following reasons: they were not available for sequencing (n = 4), the assembled genomes did not pass quality control requirements (n = 4), a discrepancy in the genetically determined genus compared with the genus as determined with MALDI-TOF (n = 1), they were clonal isolates compared with a second isolate from the same sample (n = 8) and no ESBL gene was detected in the isolate (n = 7). This resulted in 216 isolates from 346 cultured samples for

**Table 1. Prevalence of contamination with ESBL-E in retail chicken meat in the Netherlands according to year of purchase of the sample, supermarket chain of purchase and method of farming.**

| | | ESBL-E | | GLM—Poisson (REE) univariable | | | GLM—Poisson (REE) multivariable | | |
|---|---|---|---|---|---|---|---|---|---|
| | Number of samples | Positive (n = 188) | | | | | | | |
| | n = 346 | n | % | RR | 95% CI | | RR | 95% CI | |
| Period of purchase | | | | | | | | | |
| 2014 | 142 | 97 | 68.3 | Ref | | | Ref | | |
| 2015 | 204 | 91 | 44.6 | 0.65 | 0.54 | 0.79 | 0.69 | 0.58 | 0.83 |
| Method of farming | | | | | | | | | |
| Conventional | 247 | 152 | 61.5 | Ref | | | Ref | | |
| Free range | 99 | 36 | 36.4 | 0.59 | 0.45 | 0.78 | 0.60 | 0.46 | 0.78 |
| Supermarket chain | | | | | | | | | |
| SC1 | 82 | 31 | 37.8 | Ref | | | Ref | | |
| SC2 | 83 | 37 | 44.6 | 1.18 | 0.82 | 1.70 | 1.22 | 0.87 | 1.73 |
| SC3 | 100 | 58 | 58.0 | 1.53 | 1.11 | 2.12 | 1.41 | 1.04 | 1.91 |
| SC4 | 81 | 62 | 76.5 | 2.03 | 1.50 | 2.74 | 2.12 | 1.60 | 2.81 |

Abbreviations: ESBL-E, extended-spectrum beta-lactamase-producing *Enterobacteriaceae*; GLM, generalized linear model; REE, robust error estimation; RR, relative risk; n, number; CI, confidence interval; Ref, reference

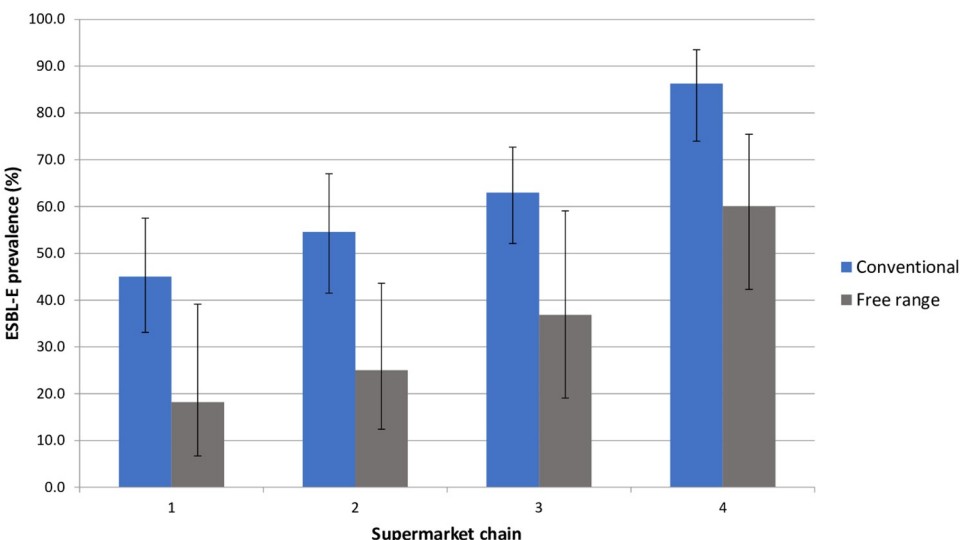

**Fig 2. The prevalence of contamination with ESBL-E according to method of farming and supermarket chain.**
Error bars show the 95% confidence intervals.

further analyses: 204 (94.4%) were *E. coli*, 11 (5.1%) were *K. pneumoniae* and one (0.5%) isolate was *E. fergusonii*. Regarding antimicrobial resistance, 51 (23.6%) isolates were phenotypically resistant to ciprofloxacin, 108 (50.0%) to norfloxacin, 111 (51.4%) to trimethoprim-sulfamethoxazole, 12 (5.6%) to tobramycin, 14 (6.5%) to gentamicin, 1 (0.5%) to piperacillin-tazobactam, and 18 (8.3%) to amoxicillin-clavulanic acid. No isolates were phenotypically resistant to meropenem, imipenem or colistin.

STs and phylogroups were determined for the 204 *E. coli* isolates. The most common STs were: ST117 (16.2%), ST10 (8.8%), ST602 (7.4%), ST88 (4.4%) and ST57 (3.9%), see Table 2. Frequencies of *E. coli* phylogroups were as follows: 61 (29.9%) isolates belonged to phylogroup A, 44 (21.6%) to B1, 41 (20.1%) to F, 17 (8.3%) to D, 16 (7.8%) to E, 13 (6.4%) to C, 2 (1.0%) to Clade I, 1 (0.5%) to B2 (non-ST131) and for 9 isolates (4.4%) the phylogroup was undetermined. Isolates within the same ST always had the same phylogroup, except for one isolate of ST10 where the phylogroup was undetermined. For all individual isolates the detected STs and corresponding phylogroups are given in S2 Table.

Among *K. pneumoniae* isolates (n = 11) the following STs were detected: ST231 (n = 3, 27.3%), ST1530 (n = 2, 18.2%) and one isolate (9.1%) of ST15, ST280, ST307, ST607, ST2176 and ST3161. The *E. fergusonii* isolate was determined as ST8330 with the *E. coli* scheme from Enterobase [https://enterobase.warwick.ac.uk/species/ecoli].

A total of 220 ESBL genes were detected in 216 isolates. The most common ESBL genes were *bla*$_{CTX-M-1}$ (n = 88, 40.0%) and *bla*$_{SHV-12}$ (n = 70, 31.8%). The *bla*$_{CTX-M-15}$ gene was found in five isolates (2.3%). Four isolates (*E. coli* n = 3 and *K. pneumoniae* n = 1) contained more than one ESBL gene: two isolates with *bla*$_{CTX-M-1}$ and *bla*$_{SHV-12}$, one isolate with *bla*$_{CTX-M-1}$ and *bla*$_{CTX-M-2}$ and one *K. pneumoniae* isolate contained *bla*$_{TEM-52B}$ and *bla*$_{SHV-27}$. The frequency distribution of all detected ESBL genes is given in Table 3.

Investigating all antimicrobial resistance genes from the ResFinder database resulted in hits with 62 different genes. For genes and percentage of isolates the genes were detected in, see S2 Fig. The most common antimicrobial resistance genes detected besides the aforementioned ESBL genes were: *sul*2 (n = 119, 55.1%) and *sul*1 (n = 62, 28.7%) conferring resistance to sulphonamides; *tet*(A) (n = 101, 46.8%), conferring resistance to tetracyclines; *aad*A1 (n = 81,

**Table 2. Frequency distribution of the *E. coli* sequence types (ST) and the corresponding phylogroups cultured from retail chicken meat in the Netherlands.** *13 STs were found in two isolates each and 38 STs were found only once.

| Sequence Type | No. Isolates (n = 204) | (%) | Phylogroup |
|---|---|---|---|
| 117 | 33 | 16.2 | F |
| 10 | 18 | 8.8 | A |
| 602 | 15 | 7.4 | B1 |
| 88 | 9 | 4.4 | C |
| 57 | 8 | 3.9 | E |
| 58 | 6 | 2.9 | B1 |
| 69 | 6 | 2.9 | D |
| 752 | 6 | 2.9 | A |
| 1158 | 4 | 2.0 | Undetermined |
| 1818 | 4 | 2.0 | A |
| 3778 | 4 | 2.0 | F |
| 38 | 4 | 2.0 | D |
| 665 | 4 | 2.0 | A |
| 115 | 3 | 1.5 | D |
| 155 | 3 | 1.5 | B1 |
| 162 | 3 | 1.5 | B1 |
| 189 | 3 | 1.5 | A |
| 5183 | 3 | 1.5 | A |
| 93 | 3 | 1.5 | A |
| pairs* | 26 | 12.7 | - |
| singletons* | 38 | 18.6 | - |
| undetermined | 1 | 0.5 | E |

37,5%), conferring resistance to spectinomycin and streptomycin; and *str*A (n = 66, 30,6%) and *str*B (n = 65, 30,1%), conferring resistance to streptomycin. The presence or absence of all individual antimicrobial resistance genes for all individual isolates is shown in S2 Table.

IncFIB, Col, IncI and IncFII were the most abundant plasmid replicon families with a frequency of 80.1%, 77.3%, 69.0% and 55.6%, respectively. The detected plasmid replicon families and the number of isolates in which they were detected are shown in S3 Table.

**Table 3. Frequency distribution of detected ESBL genes in ESBL-E isolates cultured from retail chicken meat in the Netherlands.** 220 detected ESBL genes from 216 ESBL-E isolates.

| ESBL gene | Frequency n = 220 (%) | *E. coli* | *K. pneumoniae* | *E. fergusonii* |
|---|---|---|---|---|
| $bla_{CTX-M-1}$ | 88 (40.0) | 88 | | |
| $bla_{SHV-12}$ | 70 (31.8) | 70 | | |
| $bla_{TEM-52C}$ | 23 (10.5) | 23 | | |
| $bla_{TEM-52B}$ | 18 (8.2) | 16 | 1 | 1 |
| $bla_{SHV-2}$ | 9 (4.1) | 2 | 7 | |
| $bla_{CTX-M-15}$ | 5 (2.3) | 4 | 1 | |
| $bla_{SHV-5}$ | 2 (0.9) | | 2 | |
| $bla_{CTX-M-2}$ | 2 (0.9) | 2 | | |
| $bla_{TEM-15}$ | 1 (0.5) | 1 | | |
| $bla_{CTX-M-32}$ | 1 (0.5) | 1 | | |
| $bla_{SHV-27}$ | 1 (0.5) | | 1 | |

## Investigation of clonality using whole-genome MLST

Clonality within the *E. coli* isolates was investigated using wgMLST, a neighbour-joining tree of the data is shown in S3 Fig. A total of 20,706 pairwise comparisons were made of which 148 (0.7%) were within the threshold of clonality. As a sensitivity analysis the cut-off value for clonality of the wgMLST was doubled to 0.019, this increased the percentage of clonality from 0.7% to 0.8%.

Most of the isolates from which the pairwise isolate comparisons indicated clonality belonged to a limited number of sequence types (ST): ST602 (n = 87, 58,8%), ST117 (n = 24, 16,2%), ST10 (n = 8, 5.4%), ST69 (n = 6, 4,1%), ST57 (n = 4, 2,7%) and ST1158, ST58 and ST88 (each with n = 3, 2.0%). There were ten other pairs of clonally related isolates, all with their own sequence type; one in which the wgMLST clonally related pair of isolates consisted of two different conventional ST, ST45 and ST8567. The frequency of clonality within sequence types was 4.5% and 5.2% for ST117 and ST10, respectively, whereas it was 82.9% for ST602, S4 Table. The median number of days between time of purchase of the samples that the clonally related isolates were cultured from was longer in ST602 (median 94 days, range 0–226 days), compared to ST117 (median 6.5 days, range 0–346 days) and ST10 (median 8 days, range 0–21 days), see S4 Table.

The general trend in frequency of clonality shows a decrease in clonality with increasing time interval between isolates as is shown in Table 4 and in more detail in S4 Fig. However, the first months show an increase in clonality with up to five months between the isolates showing

**Table 4. Frequency of clonality of the pairwise isolate comparisons and the univariable and multivariable regression analyses on the different epidemiological relations.**

| | No. clonally related comparisons | No. comparisons | % clonally related | GLM–binomial (REE) univariable | | GLM–binomial (REE) multivariable | |
|---|---|---|---|---|---|---|---|
| | | | | RR | 95% CI | ARR | 95% CI |
| Time between isolates | | | | | | | |
| 0–6 months | 123 | 8973 | 1.37 | ref | | ref | |
| 6–12 months | 24 | 6144 | 0.39 | 0.29 | 0.18–0.44 | 0.29 | 0.19–0.45 |
| > 12 months | 1 | 5589 | 0.02 | 0.01 | 0.00–0.09 | 0.01 | 0.00–0.10 |
| Method of farming | | | | | | | |
| Between | 35 | 6308 | 0.55 | ref | | ref | |
| Within | 113 | 14398 | 0.78 | 1.41 | 0.97–2.06 | 1.40 | 0.96–2.03 |
| Supermarket chain | | | | | | | |
| Between | 83 | 15161 | 0.55 | ref | | ref | |
| Within | 65 | 5545 | 1.17 | 2.14 | 1.55–2.96 | 2.02 | 1.47–2.79 |
| Individual supermarket chain comparisons | | | | | | | |
| SC3 | 29 | 1891 | 1.53 | | | | |
| SC3/SC4 | 57 | 4278 | 1.33 | | | | |
| SC4 | 26 | 2346 | 1.11 | | | | |
| SC1 | 5 | 528 | 0.95 | | | | |
| SC2 | 5 | 780 | 0.64 | | | | |
| SC2/SC3 | 10 | 2480 | 0.40 | | | | |
| SC2/SC4 | 9 | 2760 | 0.33 | | | | |
| SC1/SC2 | 2 | 1320 | 0.15 | | | | |
| SC1/SC4 | 3 | 2277 | 0.13 | | | | |
| SC1/SC3 | 2 | 2046 | 0.10 | | | | |

Abbreviations: No., number of; GLM, generalized linear model; REE, robust error estimation; RR, relative risk; ARR, adjusted relative risk; CI, confidence interval.

the highest rates of clonality. All clonally related isolates with 3–5 months between the isolates belong to ST602, see S5 Table for frequency of clonality per month per ST. No clonal related-ness was found in isolates more than 13 months apart. The frequency of clonality within super-market chains was higher than the frequency of clonality between supermarket chains, see Table 4. This holds true with the exception of supermarket chain 3 and supermarket chain 4, for which the between supermarket chain frequency of clonality was higher than most other within supermarket chain comparisons. See Table 4 for all individual supermarket chain com-parisons. No effect on the frequency of clonality within or between methods of farming was observed. In the multivariable analyses within supermarket chain comparisons were twice as likely to be clonally related compared with between supermarket chain comparisons, adjusted RR of 2.0 with 95% CI 1.5–2.8. As time intervals were chosen with knowledge of the data, dif-ferent models were tested, see S6 Table. Decreased time intervals of four months between the isolates in the first year showed higher clonality with 5–8 months between isolates compared to 1–4 months between the isolates. This was however followed by the expected decrease in clonality. Also, the time component was removed from the multivariable analyses. These changes to the model had little impact on the point estimates for supermarket chain or method of farming.

## Discussion

In the current study the prevalence of contamination with ESBL-E in retail chicken meat was investigated over a period of two years in the Netherlands. First, a decrease in prevalence of contamination with ESBL-E was seen over time. Second, the method of farming was associated with the prevalence of contamination with ESBL-E; free-range chicken meat had a lower ESBL-E prevalence compared with conventional chicken meat. Third, the ESBL-E prevalence in retail chicken meat differed between supermarket chains. These three factors were all inde-pendently associated with the prevalence of contamination with ESBL-E in a multivariable model.

Two datasets have been described in peer-reviewed literature on the presence of ESBL-E in retail chicken meat in the Netherlands. Cohen Stuart *et al*. and Leverstein van Hall *et al*. reported an ESBL-E prevalence of 94% (tested samples: 98) in chicken meat purchased in 2010 [12,13]; and Overdevest *et al*. reported a prevalence of 79.9% (tested samples: 89) in randomly chosen packages of retail chicken meat purchased in 2009 [14]. Yearly updates on antimicro-bial use and resistance data in the veterinary sector are published in the Netherlands in the "Monitoring of Antimicrobial Resistance and Antibiotic usage in Animals in the Netherlands (MARAN)" reports [20,44–46]. These reports also describe the ESBL- and/or AmpC- (ESBL/AmpC) producing *Enterobacteriaceae* prevalence in retail chicken meat. The reported results by MARAN are not directly comparable with the current study as the culture methods are dif-ferent. Also, besides ESBL-E, AmpC-producing *Enterobacteriaceae* are included in the reported numbers. Despite these differences, the decrease in ESBL-E prevalence in retail chicken meat is similar in the MARAN reports compared to the current study. Confirmed ESBL/AmpC-producing *Enterobacteriaceae* were present in 73% and 83% of tested samples in 2012 and 2013, respectively [46,47]. This was followed by a decrease in 2014 and 2015, with the lowest prevalence (24%) of ESBL/AmpC producing *E. coli* reported in fresh chicken meat in 2016; which increased again in 2017 to 31.6% [20,44]. A decreasing ESBL-E prevalence was also reported from broiler faeces, both in selective cultures for ESBL/AmpC producing *E. coli* and in the proportion of cefotaxime resistance in non-selectively cultured *E. coli* isolates [44].

Different articles have reported on the effect of farming practices on antimicrobial resistant microorganisms in meat products. Cohen Stuart *et al*. found high ESBL-E prevalence both in

conventional, 100% (95%CI 92.8% - 100.0%) and organic chicken meat, 81.6% (95%CI 66.3–91.1%) [12]. In a study by Miranda *et al.* that looked at resistance rates to eight different types of antibiotics in randomly picked *E. coli* isolates, the resistance rates were higher in conventional chicken meat compared to organic chicken meat [48]. Looking at resistance rates in randomly selected *E. coli* isolates to 12 types of antibiotics, under which three cephalosporin's, Davis *et al.* found differences in resistance rates in turkey meat with different antibiotic use claims, but found that in chicken meat the brand of the meat had a larger effect than the antibiotic use claim [49]. The current study finds effects of both the supermarket chain and the method of farming used. Free-range chickens receive less antibiotics compared with conventionally farmed chickens, which could be a factor related to this observed difference [50].

Comparing the ESBL-E genes detected on retail chicken meat from the current study with previously published data shows broadly similar results with $bla_{CTX-M-1}$ being the dominant gene [18,51]. Other genes frequently present are $bla_{SHV-12}$, $bla_{TEM-52B}$ and $bla_{TEM-52C}$. In the current study the frequency of $bla_{SHV-12}$ is higher compared to the numbers found in retail chicken meat as described in the aforementioned study [18,51]. This may be due to differences in culture techniques (MARAN does not use selective plates with ceftazidime in addition to selective plates with cefotaxime) or to differences in sampling, for instance from a supermarket chain not included in the current study. Another difference is the high frequency of $bla_{CTX-M-2}$ in meat samples in 2014 described by MARAN, which in that report was comparable to the frequency of $bla_{CTX-M-1}$ [45]. In the same year the current study did not detect any $bla_{CTX-M-2}$ and it was only sporadically detected in June 2015. We currently have no explanation for this difference. The $bla_{CTX-M-15}$ gene, which is the most frequently detected ESBL gene in human infections in the Netherlands, was detected in 2.3% of the isolates [14,18,52].

The most abundant STs from the current study, ST117 and ST10 are in concordance with previously published data from chicken meat in the Netherlands. [13,14] The third most common sequence type, ST602 has not been described in Dutch poultry to the best of our knowledge, but has frequently been described in poultry in other countries such as Sweden, Japan, England and Tunisia [18,53–56].

Clonal relatedness of the *E. coli* isolates from the current study was investigated using a cut-off for clonal relatedness that was set to determine clonal spread within a hospital setting in a timeframe of 30 days [27]. The current study has a different setting, with potential epidemiological relations more distant compared to that for which the cut-off was set, thus less stringent cut-off values were considered. Doubling the cut-off value for the wgMLST only had a small effect on the frequency of clonality. As such, the original cut-off value was used.

We were surprised by the relation of time between the isolates and the frequency of clonality. We expected a decrease over time, but found an increase in frequency of clonality up to five months between the isolates. After this increase in frequency of clonality it declines rapidly with the maximum time between clonally related isolates being 13 months. The increase in the frequency of clonality in the first months is almost solely caused by ST602. This highly clonal cluster stands out and the clonally related isolates have a longer median time between isolates compared to other clusters (ST10 and ST117). A possible explanation could be relatively low genetic variability in the sequence type. However, continued introduction to the food chain from a point source or temporary storage of a contaminated batch are other possibilities to explain the observation. Different options to cope with this time observation were tried in multivariable models that also looked at the effect of the supermarket chain and the method of farming on the frequency of clonality. In the different models the effect sizes of the latter two factors remained stable but the effect size of time between the isolates fluctuated with the different categorical options for time. We believe the key message on time between the isolates and clonality is that almost no clonality is seen in samples more than 12 months apart.

The second message from the clonality analysis is that isolates are twice as likely to be clonally related when the isolates are from within one supermarket chain, compared to isolates from different supermarket chains. This may be explained by overlapping production chains that give rise to more epidemiological relations between the isolates. Such relations could be isolates from chicken meat from the same farm, or possible contaminations from a common source in the processing of the meat. The higher frequency of clonality between supermarket chain 3 and supermarket chain 4 suggests a common source somewhere in the production chains. Clustering isolates closely matched in time could be due to batch contamination during processing of the meat, transmission between chickens or the chickens acquired the isolates from a common source. Clonal isolates cultured from samples collected months apart could have a wide range of possible sources of contamination. We could not verify hypotheses of where contamination or transfer may have occurred, as the production chain of the individual chicken meat samples was not accessible to us. However, combining this type of high-resolution typing data with precise knowledge of the flow of the products through the production chain and the possibility to go back and sample through that production chain could create the possibility to eliminate steps where contamination of meat products occurs.

Strengths of the study are the focus on a specific and frequently used product, raw chicken breast fillet. Choosing one specific type of product allowed investigations into differences between free-range and conventional chicken meat and differences between supermarket chains. Carefully performed sampling, including only one sample per supermarket chain per day (or with different batch numbers), to minimize possible effects of batch contaminations on the prevalence of contamination with ESBL-E and relative gene abundance. A selective pre-enrichment step and a well-tested ESBL-E screening agar was used to ensure a high sensitivity in detecting ESBL-E in the samples [57,58]. The use of WGS enabled molecular detection of all currently known genes responsible for an ESBL phenotype. It also allowed for genetic screening of resistance genes other than ESBL, ST identification and phylotyping. In addition, WGS will allow future genetic evaluations as time passes and future comparisons with other strain collections.

The current study gives a precise genetic overview of the ESBL genes and isolates found in chicken breast fillet, a broader selection of chicken products may have increased the variability of the gene content. Secondly, although care was taken during sampling to obtain a good representation of chicken breast fillet over time, it would have been preferable to have had a more continuous sampling strategy instead of periods with higher intensity sampling and different periods, including the last four months of 2015, with no samples being taken. A third limitation of the study is the fact that no quantitative cultures were performed on the chicken meat. Therefore, we cannot conclude on the bacterial (ESBL-E) load per sample over time. A final point of caution is the fact that the prevalence of ESBL-E has been known to vary over time and the timeframe in the current study is relatively short. However, the measured decrease in ESBL-E prevalence is considerable and the prevalence of ESBL/AmpC-producing Enterobacteriaceae has been shown to remain low. The MARAN reports show rates of contamination of retail chicken meat of 24% and 31.6% in 2016 and 2017 respectively which supports that our findings indicate a sustainable reduction of ESBL-E in retail chicken meat [20,44].

Chicken meat is a frequently consumed product and is known to often be contaminated with ESBL-E. Combining these facts, retail chicken meat is a potential source of ESBL-E for humans. Better understanding of factors that describe the prevalence of contamination with ESBL-E creates opportunities for concrete control measures and allows for a more in-depth analysis of production chains. What also makes the data from the current study relevant is that much effort was made to decrease antibiotic use in veterinary sector starting from 2009. The

total decrease in antibiotic sales for the complete veterinary sector was 58% from 2009 to 2014 [20]. The antibiotic sales were relatively stable during the time frame of the sample collection for the current study [20]. Although it is interesting that in the years after a large decrease in antibiotic use in the veterinary sector the prevalence of contamination with ESBL-E on retail chicken meat subsequently decreased, no conclusions on the possible causality of these observations can be drawn. The study design was not intended to look at this relation and there are too many unknown factors that could also influence the ESBL-E prevalence on retail chicken meat such as changes in the slaughter process, changes in packaging practices and differences in the origin of the meat sold in the supermarkets.

Concluding, the current study describes a decreasing prevalence of contamination with ESBL-E in retail chicken meat in the Netherlands from December 2013 until August 2015. The prevalence of contamination with ESBL-E was lower in free-range chicken meat compared with conventional chicken meat and also varied between supermarket chains. In pairwise isolate comparisons, clustering occurs more often within supermarket chains than between supermarket chains and clustering was not found in isolates cultured more than 13 months apart.

## Supporting information

**S1 Table. Number of samples taken per month, per supermarket chain and per method of farming.**
(DOCX)

**S2 Table. Table showing species, multilocus sequence type, *E. coli* phylogroup, day of sampling after start of study, anonymized supermarket chain and the presence and or absence of the different resistance genes and results of phenotypic antimicrobial susceptibility testing for all isolates in the study.**
(XLSX)

**S3 Table. Detected plasmid replicon families and the number of ESBL-E isolates from retail chicken meat they were detected in.**
(DOCX)

**S4 Table. Frequency of clonality within the three most common multilocus sequence types with the median, minimum and maximum time between isolates for related and unrelated isolates within the ST.**
(DOCX)

**S5 Table. Number of clonally related isolate comparisons per sequence type (ST) with increasing time between isolates, in months.** Within the pairwise comparisons all ST were the same except for an isolate with ST43 which was clonally related to an isolate of ST8567.
(DOCX)

**S6 Table. Alternate multivariable models of frequency of clonality using wgMLST, excluding the time variable in alternative model 1 and using shorter time periods in the first year in alternative model 2.**
(DOCX)

**S1 Fig. Prevalence of contamination with ESBL-E in free range and conventional retail chicken meat in the Netherlands.** X-axis shows time in months after start of the study, error bars show 95% confidence intervals.
(TIF)

**S2 Fig. Prevalence of ESBL genes and other genes associated with antimicrobial resistance in ESBL-E isolates cultured from retail chicken meat.** Abbreviations: Sulfon, sulfonamide; Phen, phenicol; Li, lincosamide; Fo, fosfomycin; ESBL, extended-spectrum beta-lactamase; AQ, aminoglycoside and quinolone.
(TIF)

**S3 Fig. Neighbour-joining tree based on the wgMLST analysis of 204 ESBL-producing *E. coli* isolates cultured from retail chicken meat in the Netherlands, using a "pairwise ignore missing values" approach.** Legend, circles from inside out: conventional sequence type; shading in light or dark grey of the sequence type indicates clustering in whole genome multilocus sequence typing analyses; method of farming, light green is conventional and dark green is free range; supermarket chains: red SC4, light-green SC3, purple SC2, grey-cyan SC1; the outer most ring shows the detected ESBL genes in each isolate.
(TIF)

**S4 Fig. Frequency of clonality with increasing time between the samples from which the isolates were cultured.**
(TIF)

## Acknowledgments

This publication made use of the PubMLST website (https://pubmlst.org/) developed by Keith Jolley and sited at the University of Oxford [59]. The development of that website was funded by the Wellcome Trust.

## Author Contributions

**Investigation:** Pepijn Huizinga, Marjolein Kluytmans-van den Bergh, Ina Willemsen, Carlo Verhulst, Silvia García-Cobos, Jan Kluytmans.

**Writing – original draft:** Pepijn Huizinga, Jan Kluytmans.

**Writing – review & editing:** Pepijn Huizinga, Marjolein Kluytmans-van den Bergh, John W. Rossen, Ina Willemsen, Paul H. M. Savelkoul, Alexander W. Friedrich, Silvia García-Cobos, Jan Kluytmans.

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
