## [Decision Letter · Decision Letter 0]

26 Sep 2019

PONE-D-19-22286

Decreasing prevalence of contamination with extended-spectrum beta-lactamase-producing Enterobacteriaceae (ESBL-E) in retail chicken meat in the Netherlands

PLOS ONE

Dear Mr. Huizinga,

Thank you for submitting your manuscript to PLOS ONE. After careful consideration, we feel that it has merit but does not fully meet PLOS ONE’s publication criteria as it currently stands. Therefore, we invite you to submit a revised version of the manuscript that addresses the points raised during the review process.

Two reviewers assessed your manuscript and brought forth important points that need to be addressed before the manuscript can be accepted for publication. 

We would appreciate receiving your revised manuscript by Nov 10 2019 11:59PM. To enhance the reproducibility of your results, we recommend that if applicable you deposit your laboratory protocols in protocols.io, where a protocol can be assigned its own identifier (DOI) such that it can be cited independently in the future. For instructions see: http://journals.plos.org/plosone/s/submission-guidelines#loc-laboratory-protocols

We look forward to receiving your revised manuscript.

Kind regards,

Timothy J. Johnson

Academic Editor

PLOS ONE

Journal Requirements:

2. In your Methods section, please provide additional details regarding how you chose the sample size for this study, e.g. power calculation.

Reviewers' comments:

Reviewer's Responses to Questions

**Comments to the Author**

1. Is the manuscript technically sound, and do the data support the conclusions?

Reviewer #1: Partly

Reviewer #2: Partly

2. Has the statistical analysis been performed appropriately and rigorously? 

Reviewer #1: No

Reviewer #2: Yes

3. Have the authors made all data underlying the findings in their manuscript fully available?

Reviewer #1: No

Reviewer #2: Yes

4. Is the manuscript presented in an intelligible fashion and written in standard English?

Reviewer #1: Yes

Reviewer #2: Yes

5. Review Comments to the Author

Reviewer #1: The study of Huizinga et al. deals with the investigation of chicken meat concerning ESBL-producing Enterobacteriaceae in the Netherlands. Samples were conducted during two different periods and respective isolates were investigated using whole genome sequencing and wgMLST analyses. Furthermore, some statistics about possible phylogentic relationships were carried out. From their data the authers concluded a reduction in the prevalence of the resistant bacteria from 2014 to 2015 and found lower prevalence in meat from free-range chicken than in meat from conventional farms.

The laboratory methods used in the study are state of the art; however, there is a huge lack in information about the samples and the investigated "supermarket-chains" and, therefore, conclusions might be doubthly

1) the first sampling period started in December 2013 but the authors analysed 2014 to 2015. Did they included 2013 in the 2014 sample set?

2) it is discussed that two different periods were sampled (11 month vs 3 month). How many samples were taken each month? how many samples were from conventional (each month) and from free-range (each month)? this information is important to know if you really can conclude a reduction from sampling such different time periodes.

3) How are the supermarket chains defined? Where did the chain started? At the farm? The slaughterhouse? The packaging? (A graphical overview would be helpful especially as it was stated that two chains were somehow combined) How many samples each chain were taken?

4) From Figure2 one would assume each chain comprises conventional and free-range products. It´s not likely that free-range and conventional were fattened at the same farm. Were exactly the same chains with comparable numbers of samples investigated during both periods?

S1 Fig) This figure is highly missleading! Are the samples of 2013 included as well? And summarizing 3 month in two qaters (q6 and q7) indicate a much larger sampling periode than conducted. This should be changed to month and should also differentiate between free-range and conventional

S3 Fig) Legend/explanation of colors are missing

S4 Fig) what about the impact of the supermarket chain and the fattening conditions on the clonality? One would expect closer relations in the same chain. Therefore, samples should be analysed differently as indicated in table 4 and this figure can be excluded

Reviewer #2: Overall, the manuscript is interesting, extremely relevant and discussion is very well-balanced with limitations included. Please consider the following suggestions to make this manuscript even stronger.

Major comments:

Methods section:

Sample collection- Needs more epidemiological information. Are these supermarkets representative of the whole Netherlands? How geographically dispersed are these supermarket chains? Since these are chains, I assume there were multiple stores. How many stores per supermarket chain sampled? Two supermarket chains merged during study. So, were there 4 or 5 supermarkets to begin with? How was the sampling designed- random, convenient etc.?

Best before date was available but not used in any statistical analyses- would that have potentially confounded results e.g. samples closer to best before date might have higher levels of bacterial contamination.

What do you mean by “biological chicken meat”?

Microbiological methods: Did these EbSA plates contained cefotaxime or ceftazidime or were these plates split into 2 halves with one half containing cefotaxime and other half containing ceftazidime? If indeed, these plates carried just cefotaxime or just ceftazidime, then there is a bias in methods as using only one of these antimicrobials for ESBL screening can lead to differing results and false negatives.

Whole genome sequencing and quality control: Please include the read lengths, chemistry versions and instrument type for MiSeq.

Statistical analyses: For analysis described in lines 187-194: There is a huge gap in time between October 2014 and June 2015 when sampling was not done. How was this missing time accounted for in the analyses? For example, if a clonal isolate was found in June 2014 and again in June 2015, then will the difference between these clones be considered as 12 months whereas there is a possibility that a similar clonal isolate was present but not sampled during this huge time-gap in sampling.

Also, in lines 193-194: It is suggested that categories of time were chosen to coincide with frequency of clonality. I wonder if these models were built a-priori or on the basis of some statistical criteria. If models were built after looking at the data, then there might be a chance that models might be biased. Were there any sensitivity analyses performed e.g decreasing the periods from six months to 4 months or dropping some data and refitting the models to check for consistency of the results under various assumptions?

Results:

Please provide raw data (could be in supplementary form) about how many samples were collected per month, how many samples were collected per supermarket chain etc.

Table 1: How were the sample collected during December 2013 fit in this model? Are these included with 2014 isolates?

Table 2: there is no ST02 mentioned in the text. Is it ST602?

Table 3: Maintain consistency in gene nomenclature: bla should be italicized and CTX-M-1 should be a subscript.

Lines 286-287: Cite the actual median number of days and reference to the supplementary material.

Discussion: The biggest issue with both results and discussion section is the emphasis on decreasing prevalence of ESBL-E based on. A very short time period of 2 years. It has been noticed in several time-trend analysis that the prevalence of antimicrobial resistances vary (suddenly increase or decrease) considerably over short periods of time and sometimes without any apparent reasons; and longer time periods of 3-5 years are better to make any definitive claims. Hence, the authors are suggested to be include some verbiage to further highlight this limitation.

Lines 317-319: Consider restructuring the sentence beginning with “ Since a number…”

Lines 319-323: Meaning of the sentence starting with “Although the selective…” not clear at all

Lines 332-334: Did these “eight type of antibiotics” included cephalopsorins?

Line 334- “With similar methods to previously mentioned study”. Which study is being referred to? Please cite.

Lines 364-366: These sentences are critical as they imply other models were used using different categories for time. Please include these models in supplementary section.

Minor comments:

Line 40: Meaning of the sentence not clear due to use of “evaluable”

Line 61: Should be “were” instead of “was”

Lines 64-66: For the sentence starting with “The epidemiology has…” – consider restructuring the structure. Not clear whether the epidemiology of pathogens have changed or related infections have changed

Lines 66-68: For sentence starting with “Research efforts…”- consider restructuring the sentence. Routes of transmission of what- pathogens? Genetic elements?

Lines 69-70: For sentence starting with “ESBL-E contamination…”- cite the year when these rates were estimated.

Line 100: should “and/or” instead of “and or”

Lines 416: should be “decreased” instead of “decreases”

6. PLOS authors have the option to publish the peer review history of their article (what does this mean?). If published, this will include your full peer review and any attached files.

Reviewer #1: No

Reviewer #2: Yes: Shivdeep Singh Hayer

---

## [Author Response · Author response to Decision Letter 0]

1 Nov 2019

Dear Mr. Johnson, 

We kindly thank you and the reviewers for the critical appraisal of our manuscript and the constructive comments. We have tried to comply with the suggested changes and hope that with these changes you will consider the manuscript for publication in PLOS ONE. We have addressed each comment separately in the order we received the comments. This document is attached and named "Response To Reviewers" 

Kind regards, 

Pepijn Huizinga

---

## [Decision Letter · Decision Letter 1]

9 Dec 2019

Decreasing prevalence of contamination with extended-spectrum beta-lactamase-producing *Enterobacteriaceae* (ESBL-E) in retail chicken meat in the Netherlands

PONE-D-19-22286R1

Dear Dr. Huizinga,

We are pleased to inform you that your manuscript has been judged scientifically suitable for publication and will be formally accepted for publication once it complies with all outstanding technical requirements.

However, please consider the suggestions of reviewer #1 regarding use of figure S1.

With kind regards,

Timothy J. Johnson

Academic Editor

PLOS ONE

Additional Editor Comments (optional):

Reviewers' comments:

Reviewer's Responses to Questions

**Comments to the Author**

1. If the authors have adequately addressed your comments raised in a previous round of review and you feel that this manuscript is now acceptable for publication, you may indicate that here to bypass the “Comments to the Author” section, enter your conflict of interest statement in the “Confidential to Editor” section, and submit your "Accept" recommendation.

Reviewer #1: All comments have been addressed

Reviewer #2: All comments have been addressed

2. Is the manuscript technically sound, and do the data support the conclusions?

Reviewer #1: Yes

Reviewer #2: Yes

3. Has the statistical analysis been performed appropriately and rigorously? 

Reviewer #1: Yes

Reviewer #2: Yes

4. Have the authors made all data underlying the findings in their manuscript fully available?

Reviewer #1: Yes

Reviewer #2: Yes

5. Is the manuscript presented in an intelligible fashion and written in standard English?

Reviewer #1: Yes

Reviewer #2: Yes

6. Review Comments to the Author

Reviewer #1: all comments have been addressed in an approproate manner; however I would recommend to use the figure S1 provided in the comments showing prevalences of both conventional and free-range. The authors stated that they prefer there current version of S1 as they "visually show the decrease in ESBL-E prevalence". Reviewer#2 already stated before that "It has been noticed in several time-trend analysis that the prevalence of antimicrobial resistances vary (suddenly increase or

decrease) considerably over short periods of time and sometimes without any apparent reasons; " Therefore, the graph from the comments is more convincing and believable. In concordance with this I would recommend to be careful with fitting graphs.

Reviewer #2: (No Response)

7. PLOS authors have the option to publish the peer review history of their article (what does this mean?). If published, this will include your full peer review and any attached files.

Reviewer #1: No

Reviewer #2: Yes: Shivdeep Singh Hayer

---

## [Editor Report · Acceptance letter]

18 Dec 2019

PONE-D-19-22286R1 

Decreasing prevalence of contamination with extended-spectrum beta-lactamase-producing *Enterobacteriaceae* (ESBL-E) in retail chicken meat in the Netherlands 

Dear Dr. Huizinga:

I am pleased to inform you that your manuscript has been deemed suitable for publication in PLOS ONE. Congratulations! Your manuscript is now with our production department. 

With kind regards,

on behalf of

Dr. Timothy J. Johnson 

Academic Editor

PLOS ONE